# The Bioengineering of Insect Cell Lines for Biotherapeutics and Vaccine Production: An Updated Review

**DOI:** 10.3390/vaccines13060556

**Published:** 2025-05-23

**Authors:** Michał Sułek, Agnieszka Szuster-Ciesielska

**Affiliations:** Department of Virology and Immunology, Institute of Biological Sciences, Maria Curie-Skłodowska University, Akademicka 19, 20-033 Lublin, Poland

**Keywords:** insect cell lines, baculovirus expression vector systems, genetic engineering, recombinant proteins, virus-like particles, vaccines

## Abstract

Insect cell lines are a cornerstone of recombinant protein production, providing a versatile platform for biopharmaceutical and research applications. In the early 20th century, scientists first attempted to culture insect cells in vitro, developing continuous cell lines to produce the first insect cell-derived recombinant protein, IFN-β. Initial successes, along with advancements in the use of insect cells for recombinant protein manufacturing, primarily relied on baculovirus expression vector systems (BEVSs), which enable heterologous gene expression in infected cells. Today, growing attention is focused on baculovirus-free systems based on the transfection of insect cells with plasmid DNA. This approach simplifies the final product purification process and facilitates the development of stable monoclonal cell lines that produce recombinant proteins or protein complexes, particularly virus-like particles (VLPs). Thanks to advancements in genetic engineering and the application of adaptive laboratory evolution (ALE) methods, significant strides have been made in overcoming many limitations associated with insect cell BEVSs, ultimately enhancing the reliability, yield, and quality of the biomanufacturing process. Our manuscript discusses the history of developing insect cell lines, presents various recombinant protein production systems utilizing these cells, and summarizes modifications aimed at improving insect cell lines for recombinant protein biomanufacturing. Finally, we explore their implications in pharmaceutical production, particularly on Nuvaxovid^®^/Covovax, which is the latest approved vaccine developed using insect cell BEVSs for protection against SARS-CoV-2.

## 1. Introduction

Cell culture involves the growth and maintenance of live cells outside their natural environment in controlled conditions. This term applies to microorganisms and higher animals, with the latter significantly impacting modern medical biotechnology [1]. In this context, animal cell cultures can be described as factories for biopharmaceuticals. They have been used to produce various therapeutic proteins, including monoclonal antibodies, recombinant peptides, and vaccines [2]. Among the animal cell lines utilized in recombinant protein studies, insect cell cultures play a crucial role. They provide a balanced option, considering the advantages and disadvantages of various recombinant protein systems, including multiple microorganisms and higher eukaryotic platforms. While microbial techniques offer the most cost-effective and efficient methods for obtaining recombinant proteins, they have a significant drawback: the lack of substantial post-translational modifications (PTMs) present in mammalian systems [3]. Various types of yeast, which perform better in PTMs than bacteria, can execute many of these modifications similarly to mammalian systems. However, they often exhibit hypermannosylation during N-linked glycosylation, which is a critical PTM necessary for protein activity [4]. This abnormal N-glycosylation pattern may reduce human-derived protein activity or provoke unnecessary immune system responses [5]. Insect cell lines offer a wide range of PTMs, essential for determining proteins’ functional activity and diversity. One significant challenge currently being addressed is enhancing the efficiency of insect N-glycosylation to resemble the human glycoprotein pattern more closely. In this review, we examine the role of insect cell lines in recombinant protein manufacturing and present various systems based on baculovirus expression vector systems (BEVSs) and transient plasmid transfection approaches. We summarize the advantages and disadvantages of working with insect cells, emphasizing both established and novel innovations in their application for recombinant protein production. Additionally, we explore their applications in vaccine and therapeutic research, highlighting recent developments in clinical trials for pharmaceuticals developed using insect cell BEVSs.

## 2. History of Insect Cell Culture Development and Their Initial Utilization in Protein Recombination Studies

The beginning of animal cell culture is usually attributed to the first successful procedure for maintaining living cells in vitro, which was achieved by Ross G. Harrison [6]. Harrison successfully cultured frog embryo tissues from medullary tubes suspended in a drop of adult frog lymph fluid using the “hanging drop” technique, which he adapted from Koch’s work on *Bacillus anthracis* cultivation [7]. In aseptic conditions, Harrison observed the development of nerve fibers from isolated tissues for up to four weeks. Establishing a reliable method for maintaining cells outside the host’s body opened an important new chapter in biology, inspiring others to promote research on higher animals and invertebrates, including insects [8,9].

Goldschmidt was the first to successfully adapt Harrison’s culture technique for cultivating insect cells. He observed the growth and maturation of sperm cells isolated from *Samia cecropia* moths. The researcher cultured the cells in the insects’ hemolymph and kept them alive for up to three weeks [10]. This breakthrough initiated future research on spermatogenesis rather than mitosis and cell proliferation [11]. Moreover, the lack of a proper medium composition significantly limited the potential for long-term sterile cell cultivation. At that time, most media consisted exclusively of hemolymph, although hemolymph was occasionally supplemented with a mixture of salts and sugars [10,12]. Significant progress in optimizing the content of cell media was made by Trager, who developed an artificial medium in which he successfully cultivated the ovarian cells of *Bombyx mori* for up to three weeks [13]. Notably, he also succeeded in the in vitro multiplication of grasserie and equine encephalomyelitis viruses in *B. mori* and *Aedes aegypti* tissues using the artificial media he created [14]. However, the absence of antibiotics in his media made it challenging to maintain long-lasting cultures due to bacterial contamination. In 1956, Wyatt introduced a synthetic medium designed to resemble *B. mori* hemolymph. Her medium contained over twenty amino acids, five inorganic salts, three different sugars, and four organic acids. To date, it was the most complex medium, advanced both in terms of its composition and in providing optimal conditions for *B. mori* cells to grow, divide, and thrive [15].

Despite efforts, researchers still could not achieve the continuous growth of cell lines until 1959, when Gaw et al. [16] published their results on the successful maintenance of *B. mori* cells for up to 22 passages, marking the first documented monolayers used in grasserie virus studies. However, the paper by Gaw et al. did not receive the recognition it deserved in the scientific community, possibly due to its affiliation with China, which, at that time, was not seen as a prominent hub for innovative insect cell research within the predominantly American–European network. Independently of Gaw, in 1962, Thomas Grace transformed the field of insect cell cultures by obtaining the first four insect cell lines that could be maintained in continuous cultures [17]. Grace immortalized *Opodiphthera eucalypti* cells using a method he referred to as “benign (or organized) neglect”. This method involved changing half of the medium every few days in cultures lasting approximately three months, characterized by minimal growth and typically destined for disposal. Eventually, after over a year of sustaining barely viable cultures, he observed a rapid increase in cell division, yielding the first polyploid insect cell lines, in contrast to the mother’s diploid cells [17,18]. In subsequent years, he successfully established a continuous cell line of *Aedes* aegypti, which facilitated more detailed research on malaria and yellow fever virus vectors [19]. Notably, before these breakthroughs, Grace developed a cell medium that he effectively utilized in his study. He enhanced Wyatt’s existing medium by adding ten B-complex vitamins, significantly improving the longevity of cultured cells [15,20]. Interestingly, his medium is still commercially available and routinely used for cultivating cell lines derived from lepidopterans. The milestones in developing insect cell cultures are illustrated in Figure 1.

Grace’s advancements in establishing a medium that facilitated the maintenance of continuous insect cell lines signaled a new era in insect sciences. In subsequent years, researchers developed new cell lines from insect families beyond Lepidoptera and Diptera, with minor modifications to existing media. Insect cell lines became invaluable tools in studies of insect physiology and served as virus vectors for exploring animal and plant diseases [27]. Many significant scientific events occurred in the 1970s during the flourishing era of insect-cell-related research. In 1971, Vail et al. [28] reported two viruses isolated from cabbage (*Trichoplusia ni*) and alfalfa (*Autographa californica*) loopers. The latter, described as a “multiply embedded virus”, was significantly more efficient in infecting the two studied looper species than the single embedded virus isolated from *Trichoplusia ni*. The ability of the *Autographa californica* multiple nucleopolyhedrovirus (AcMNPV) to effectively infect both studied pests captured the authors’ attention regarding its potential use as a biopesticide. However, AcMNPV found a more robust application in emerging recombinant protein expression technology.

The successful construction of the first DNA recombinant molecule—a hybrid of SV40 virus DNA combined with genes encoding for λ phage and the galactose operon of *E. coli* [29]—was swiftly followed by the development of recombinant bacteria containing biologically active plasmids [30], which sparked a scientific flow that ultimately led to the production of the first recombinant protein, somatostatin [31]. These groundbreaking discoveries established a new branch of biological sciences, i.e., genetic engineering. Researchers soon envisioned utilizing insect cell lines as vectors to produce recombinant proteins. To do this effectively, it was crucial to develop an efficient cell line that would (i) be highly susceptible to viral infection, (ii) easy to handle, manipulate, and scale up, and (iii) yield a large quantity of proteins. In 1977, Vaughn et al. [32] reported the establishment of two new continuous cell lines of *Spodoptera frugiperda* (Lepidoptera), IPLB-SF-21 and IPLB-SF-1254, cultivated from immature pupa ovaries. The former cell line, now referred to in the literature as Sf21, was known for a faster doubling time and high susceptibility to AcMNPV, which had previously been utilized in other studies for virus production [33]. The status of Sf21 as an effective AcMNPV vector contributed to its use for producing the first recombinant protein in insect cells, human interferon beta (IFN-β), in 1983 [34]. This achievement was made possible by extensive prior work on the viral vector by Smith et al. [35]. They observed that the polyhedrin-encoding gene, responsible for producing the protein matrix of occlusion bodies, is not essential for forming infectious extracellular virus particles. Since the function of polyhedrin extends beyond its role in virion formation, its transcription is significantly elevated even when most viral genes are silenced. By cloning the gene for IFN-β in place of the polyhedrin gene, they achieved the robust expression of the desired protein in fully infectious viruses capable of infecting and reproducing within insect cells. Another advantage of their strategy was the simplicity of the screening methods. As mutants could not form occlusion bodies, it was straightforward to distinguish successful recombinants from wild-type viruses by analyzing the nuclei of infected cell lines, which were then used for proper infection [34,35]. The use of the new baculovirus–insect cell line platform resulted in unprecedented efficiency in the recombination process, achieving an IFN-β activity level of 5 × 10^6^ U/mL compared to *E. coli* (1 × 10^4^ U/mL) and transgenic mice (5 × 10^4^ U/mL) vectors [36,37].

Furthermore, unlike bacterial vectors, insect cells can perform complex post-translational modifications, which may be vital for maintaining the function of human proteins [38]. In this context, utilizing the AcMNPV-Sf21 platform led to the production of partially glycosylated, functional, and highly active human IFN-β, highlighting the significance of the developed system in innovative genetic engineering studies. In the subsequent years, many new insect cell lines were introduced; however, only two of them were successfully utilized in protein recombination. In 1983, Smith and Cherry isolated a clone of the Sf21 line, resulting in a comparable yet more uniform cell line named Sf9, which offers improved size and monolayer formation [39]. In 1992, Wickham et al. [40] screened eight insect cell lines for producing recombinant β-galactosidase. The BTI-Tn-5B1-4 excelled in protein production among the lines examined, outperforming Sf9 and Sf21 by two-fold and five-fold, respectively. Cultivated from *Trichoplusia ni* eggs, the BTI-Tn-5B1-4 line found significant application in recombinant protein manufacturing alongside the Sf9 and Sf21 clones. The Tn 5B1-4 line, commercially known as High Five™, is now one of the most efficient platforms for producing various secreted proteins using insect cell lines [41]. To date, over 1500 continuous cell lines have been established according to the records in the Cellosaurus database [42]. Among them, approximately 90% were derived from the Lepidoptera and Diptera orders of insects [43].

## 3. Different Approaches for Recombinant Molecule Expression in Insect Cell Lines

The most popular cell lines for producing recombinant proteins are those previously described, which are derived from *S. frugiptera*: Sf9, Sf21, Sf900+ (commercially known as expresSF+), and BTI-Tn-5B1-4 (High Five™ Cells) from *T*. *ni*. All these cell lines can be rapidly scaled up for various applications. They can be adapted to both adherent and non-adherent conditions; they do not require CO_2_ supplementation and can thrive in serum-free media [44]. Suspension insect cell cultures can be maintained in sterile reusable glass flasks and passaged almost indefinitely without decreased transfection efficiency. According to Calles et al. [45], Sf9 cells that have been passaged over 100 times may produce even higher levels of recombinant protein than those passaged less than 45 times.

Additionally, the optimal growth temperature for insect cell cultures is lower than that for mammalian cells (27–28 °C), requiring a biosafety level 1 working environment [46,47]. These factors affect the accessibility of insect cell lines in terms of scaling production size and create opportunities to reduce costs. The use of insect cell lines in recombinant protein manufacturing is predominantly conducted using baculovirus vectors. It is important to note that baculoviruses have a narrow range of potential hosts. They are non-infectious and, therefore, non-pathogenic to humans, which is crucial for safety, particularly when generating therapeutic proteins [48]. BEVSs primarily rely on Bac-to-Bac technology, in which specific site transposition in specialized *E. coli* cells occurs instead of the traditional homologous recombination of baculovirus DNA with a donor plasmid in insect cells (Figure 2A) [49]. This technique for generating recombinant baculoviruses involves three major components: (i) a donor plasmid containing the gene of interest (GOI) under a strong late polyhedrin (polh) or p10 promoter; (ii) a baculovirus shuttle vector (bacmid) that includes genes necessary for replication, virulence, and the lacZ-mini-attTn7 fusion region; and (iii) a helper plasmid encoding Tn7 transposase, which facilitates site-specific recombination. Both the bacmid and the helper plasmid are housed within specialized DH10Bac *E. coli* cells. During the transformation of *E. coli* cells, the GOI located in the donor plasmid is transposed to the bacmid via a site-specific recombination event mediated by the transposase provided by the helper plasmid strain (Figure 2A). Successful transposition disrupts the *lacZ* gene, impairing the bacterial ability to produce β-galactosidase, which is an enzyme with an activity that can be quickly evaluated through blue-white screening with the chromogenic X-gal substrate. White colonies (containing the recombinant bacmid) are selected to isolate, purify, and use recombinant baculovirus DNA to transfect insect cells. Following transfection, the bacmid is expressed in insect cells, leading to the generation of recombinant baculoviruses and protein expression. The generated baculoviruses are collected and amplified in insect cells to obtain a high-titer stock of recombinant viruses, which are then used for the large-scale infection of insect cells, ultimately resulting in a greater yield of recombinant proteins (Figure 2A) [48].

There are various commercially available kits, primarily based on Bac-to-Bac technology, the development of which was found to reduce the time needed to obtain the protein of interest and increase its yield. However, using *E. coli* as part of the recombination platform system may contribute to the presence of bacterial genome sequences. In the case of therapeutic proteins, it has been considered an essential concern since DNA contamination may affect the gene expression profile of the host [50]. Further developed systems have simplified the procedure by eliminating the presence of *E. coli* and reducing the time needed to obtain the protein of interest. In the flashBAC™ system, the bacmid genome was improved by a mutation in genes crucial to the functioning of the *naïve* virus but improving its properties as an effective vector. The most essential mutations include deletions in the *chiA* gene region encoding the viral chitinase and deletions in the *orf1629* gene region necessary for virus replication in insect cells. The transfection of insect cells with the donor plasmid containing the GOI and part of *orf1629* restore the gene’s function, enabling viral DNA to replicate and generate virus particles. This system eliminates the screening step, as viral replication occurs only after the successful recombination of the GOI. It increases the yield of recombinant proteins, especially those that are secreted or membrane-bound, by removing the presence of viral chitinase [51]. A similar approach, where homologous recombination occurs in insect cells, utilizes a BestBac™ baculovirus DNA vector. These systems exploit the inability of linear DNA genome fragments to replicate and infect insect cells. Linear viral DNA is circularized only after successful recombination with a plasmid donor carrying the GOI, restoring its infective properties [52]. To further mitigate the risk of the recircularization of viral DNA via mechanisms other than homologous recombination, this method was enhanced in the BestBac™ and DiamondBac™ platforms by a deletion within the *orf1629* gene, akin to flashBAC™ technology. Similarly to flashBAC™, following the introduction of the GOI into the viral DNA, *orf1629* was restored, leading to the generation of recombinant baculoviruses [51,53].

The expression of recombinant proteins in insect cell lines is not limited to the application of baculovirus-based techniques and can also be conducted through classical direct plasmid transfection (Figure 2B). Protein expression involves transfecting insect cell lines (usually Sf9 or High Five™) with a plasmid expression vector harboring the gene of interest. A common feature of commercially available vectors (i.e., pIZ/V5-His; pIB/V5-His) is the presence of the baculovirus-derived immediate early OpIE2 promoter. This promoter enables the constitutive expression of the desired protein, independent of baculovirus infection, thereby avoiding cellular lysis, which is unavoidable when using BEVSs (Figure 2). Plasmid vectors possess various antibiotic resistance genes (*Zeocin™* for pIZ; *Blasticidin*™ for pIB), which serve as selection markers during *E. coli* transformations and the generation of stably transfected cell lines (Figure 2(B1)). These commercially available vectors typically include a sequence encoding a polyhistidine tag (V5-His) and offer additional features for convenience. For instance, the pIZT/V5-His vector facilitates the easier selection of stable clones by expressing a Zeocin resistance protein fused with the green fluorescence protein (GFP). In contrast, the pMIB/V5-His vector includes the honeybee melittin secretion signal, enabling recombinant proteins to be secreted into the medium [54,55,56,57]. The transfection of insect cells with plasmid vectors is typically performed using commercially available reagents. These include polyethylenimine (PEI)-based derivatives, such as Transporter™ 5, or lipoplex formulations like TransIT^®^-LT1 and Cellfectin™ II, with Cellfectin™ II optimized explicitly for use with insect cells [58,59]. The production of a recombinant protein through direct transformation eliminates the time-consuming process of generating recombinant baculovirus and measuring its infectivity. Virus-free systems provide the quickest way to obtain a relatively high yield of the desired protein with a more straightforward purification procedure due to the production of the protein using a nonlytic plasmid, contrary to baculovirus-based methods [60]. Additionally, obtaining constitutive cell lines capable of stable expression offers an excellent platform for the long-term, scalable production of desired proteins, which can be achieved with a featured plasmid-based transfection process (Figure 2(B1)). Despite all the advantages of direct plasmid transfection, baculovirus-based systems still offer a more efficient and notably easier method for large-scale protein manufacturing, following the initial time-consuming steps of baculovirus stock generation. Furthermore, utilizing late powerful viral p10 or polh promoters in BEVSs results in the stronger expression of the gene of interest compared to the early viral-independent OpIE2 promoter [61]. However, direct plasmid transfection systems continue to be used and developed due to their simplicity, often yielding a final product comparable to lysis systems based on BEVSs [60,62].

The popularity and potential of baculovirus-based expression systems in the global trends of recombinant molecule manufacturing are evidenced by the introduction of the ExpiSf™ Expression System (ThermoFisher, Waltham, MA, USA) in 2018, which represents the latest system utilizing Bac-to-Bac technology. This system, like its mammalian counterparts in the Expi™ product line (Expi293™ and ExpiCHO™ launched in 2012 and 2015, respectively), is known for producing a more efficient product yield compared to the earlier insect cell-baculovirus-based expression systems. Protein production in the ExpiSf™ system employs a derivative of Sf9 cells—ExpiSf9 cells—that have been adapted for suspension growth and higher density in a specialized, chemically defined, and animal-origin-free medium known as ExpiSf CD [63]. With precisely defined reaction conditions, the achievement of the high reproducibility of results alongside robust recombinant protein expression is possible. This capability has led to the system being widely adopted to produce various recombinant molecules [59,64,65,66].

The application of insect cell lines in biomanufacturing also extends to the production of complex protein structures, with virus-like particles (VLPs) as an excellent example. VLPs are nanoscale structures, self-assembled from viral proteins, which resemble *naïve* virions in size and morphology. Most importantly, they lack a viral replicative genome, which renders these molecules non-infectious. However, due to their preserved viral-origin structures, they can be effectively recognized by the host’s immune system, eliciting both cellular and humoral immune responses. These properties make VLPs particularly intriguing for vaccine development because, aside from efficiently inducing an immune response, there is no risk of viral replication in host cells. In addition to their application in vaccine development, VLPs are beneficial in gene therapy and targeted medicine. These particles can be effectively loaded with peptides or small molecules and further tailored to display the epitopes of antigens, ensuring their specific delivery to desired locations [67,68]. Due to the numerous advantages of insect cell lines, BEVSs are the most commonly used systems for obtaining VLPs among other eukaryotic and prokaryotic recombinant systems. They provide the stable and relatively rapid expression of VLPs, ensuring appropriate conditions for the development of vaccines against rapidly mutating viruses, such as influenza or SARS-CoV-2 [69]. Moreover, this system provides the most efficient generation of complex multi-protein VLPs [70]. VLPs are successfully generated using various approaches, including (i) the classical Bac-to-Bac system [71,72]; (ii) a multi-gene expression system based on MultiBac technology or the co-infection of insect cells with multiple baculovirus vectors [73,74]; and (iii) the implementation of stable recombinant insect cell lines [75]. Most recently, Lampinen et al. [62] demonstrated (iv) the effective implementation of the plasmid-based production of norovirus, rotavirus, and enterovirus-LPs in insect cell lines. The removal of baculoviruses throughout the VLP generation process contributed to the simplification of the protein purification procedure without affecting the yield of the noro- and enter-VLPs, similar to the BEVS-based approach.

## 4. Strategies for Enhancement of Biomanufacturing with Insect Cell Lines

### 4.1. Reducing Cell Debris Contamination and Enhancing Glycosylation Patterns—Baculovirus Modification Techniques

Despite the many advantages of using insect cell lines over other recombinant platforms, two significant drawbacks are currently being addressed: reducing the contamination of cell debris during protein manufacturing with BEVSs and improving glycosylation patterns. Conducting protein expression with BEVSs involves the transcription of the GOI under a late-stage promoter, typically polh or p10, which often coincides with the expression of viral lytic proteins [76,77]. As a result, the recombinant protein production process destroys insect cells, causing considerable contamination of the working environment with cellular debris, virus particles, and proteins of both viral and insect origins, including proteases. This contamination presents protein purification challenges, decreasing the final product yield [47]. Strategies to address this issue focus on improving baculovirus vectors through genetic engineering to impair problematic genes, remove unnecessary genes, or enhance beneficial ones related to protein recombination. A prime example is knocking out genes encoding chitinase and v-cathepsin, i.e., hydrolytic enzymes involved in breaking down the host’s cell walls, thus facilitating the purification process [78,79,80]. This approach is utilized in commercially available kits, such as version 2.0 of Best Bac^TM^ and the MultiBac^TM^ system, which are designed for multigene expressions [81,82,83]. Recently, X. Zhang et al. [84] evaluated the knockout of various AcMNPV genes unrelated to viral replication functions. Screening studies based on the deletion of 14 DNA fragments showed that 9 of them improved the reporter protein expression. Compared to the original vector, their studies led to the development of two new baculovirus vectors with enhanced capacity for recombinant protein production. Conversely, Martínez-Solís et al. [85] demonstrated the additive properties of the pSeL promoter, which was cloned between the polh promoter and the GFP-encoding sequence. The expression of this vector construct (polh-pSeL-GFP) exhibited higher reporter protein expression than the pSeL or polh promoters alone. Another study revealed that introducing the vp39 promoter with burst sequences along the polh promoter enhances the expression efficiency of enhanced green fluorescent protein (eGFP) [86]. Interestingly, upon the inclusion of burst sequences within the p10 promoter of *Bombyx mori*, NPV inhibited its activity despite the similarities between polh and p10 as late-stage baculovirus promoters [87]. Similar genetic engineering strategies have also been applied to achieve mammalian-like protein glycosylation patterns. Insects and mammals share the typical N-glycan structure, paucimannose, characterized by branching and terminating (in the case of insects) residues of mannoses. In mammalian cells, these two branches are further elongated with N-acetylglucosamine (GlcNAc), galactose, and terminal sialic acid residues, determining the activity of many human-origin glycoproteins [88]. Initially, human glycosylation patterns were attempted by simultaneously coinfecting insect cells with different recombinant baculoviruses, some of which encoded for GOI or glycosyltransferases and were capable of extending the human N-linked glycosylation of the paucimannose structure [89,90]. However, the inconsistency of the coinfection process could not guarantee the successful recombination of a single cell with two different vectors, which could result in cells expressing only one inserted sequence of either GOI or glycosyltransferase. This issue was resolved by encoding glycosyltransferase genes into a multigene expression platform: the MultiBac system. The adaptation of the MultiBac system for enhancing glycosylation patterns was named SweetBac and was successfully implemented for recombinant protein expression with humanized-like N-glycans in insect cells [91,92,93].

### 4.2. Enhancing the Survival of Insect Cell Cultures Following Baculovirus Infection

Genetic engineering strategies extend beyond baculovirus vectors; they are also effectively used in insect cell lines to enhance protein recombination efficiency and replicate human glycosylation patterns. A key method for increasing protein yield involves prolonging the lifespan of insect cells after viral infection. This can be achieved by introducing anti-apoptotic genes or disabling pro-apoptotic genes within the host genome. One of the anti-apoptotic genes employed in the genetic engineering of insect cell lines is the *P35* gene, which is encoded by AcMNPV and the *B. mori* nuclear polyhedrosis virus [94]. Cartier et al. [95] first demonstrated that Sf21 cell lines, when stably transfected with AcMNPV *P35*, exhibited higher resistance to apoptosis induced by chemicals or baculoviruses. Later, Lin et al. [96] successfully developed an Sf9 cell line that stably expressed P35, which showed increased resistance to apoptosis induction from nutrient stress and produced higher levels of recombinant protein than wild-type Sf9 cells (Figure 3).

Another group of proteins found to prolong cell survival includes viral ankyrins (vankyrins). Fath-Goodin et al. [97] demonstrated that incorporating the *Vank-1* gene derived from *Campoletis sonorensis* polydnavirus enhanced the survival of Sf9 cells following baculovirus infection. The transgenic clones of Sf9 cells, stably expressing the P-vank-1 protein, exhibited strong inhibitory properties toward caspase-3 activity and the internucleosomal degradation of genomic DNA after chemical or UV-induced apoptosis. The anti-apoptotic properties of the P-vank-1 protein ultimately extend the survival of baculovirus-infected cells, significantly increasing the yield of recombinant products [97,98]. An increased recombinant protein yield was also observed in other cell lines stably expressing vankyrin proteins: *Trichoplusia ni* High Five™ and SfSWT-4 [99]. Furthermore, when these vankyrin-expressing cell lines were infected with a vankyrin-enhanced baculovirus vector, a synergistic effect resulted in the elevated production of various recombinant proteins compared to infections with control viruses lacking the vankyrin gene (Figure 3) [99].

Another approach to prolonging the longevity of insect cells is the impairment of pro-apoptotic genes, which is commonly performed using RNA interference technology. In the context of increasing the efficiency of recombinant protein production using insect cell lines, this technology was used for the first time by March et al. [100]. The authors successfully silenced the *TSC1* gene in *Drosophila* S2 cells using double-stranded RNA, resulting in an increased cell growth rate and enhanced reporter (GFP) recombinant protein synthesis. This strategy was effectively adapted to impair caspase-1 activity in various Lepidopteran cell lines, including High Five^™^ [101], Sf9 [102], and the *Bombyx* mori-derived cell line BmN [103]. In Lepidoptera, this enzyme serves as a central effector caspase (analogous to human caspases-3/-6/-7) capable of triggering the final step of the apoptosis cascade upon receiving appropriate stimuli from the initiator caspase [101,102,103]. Silencing caspase-1 resulted in nearly a two-fold increase in the recombinant product yield in High Five^™^ and Sf9 and an even more significant increase in BmN cell lines [101,102,103]. All three cell lines mentioned were established as stable clones with the suppressed expression of caspase-1, eliminating the need for further double-stranded RNA treatment (Figure 3). This demonstrates the advantage of this technology in developing new production-oriented cell lines with the knockout of desired genes.

### 4.3. Improving the Ease of Working with Insect Cell BEVSs

The primary goal of the genetic engineering of insect cell lines is to enhance their longevity, thereby increasing the overall product yield. However, various strategies to improve insect cell lines aim to shorten the entire protein generation process. The expression of recombinant molecules using baculovirus-based platforms involves time-consuming steps for generating and validating the infectivity of viral stock (Figure 2). Modern methods for determining virus quality employ molecular techniques like real-time PCR or flow cytometry, which enable rapid and accurate viral load assessments [63,104,105]. However, these methods require sophisticated equipment, costly reagents, and numerous optimization steps.

Furthermore, there is usually no clear distinction made between active and inactive virus particles. In real-time PCR, the gathered data are based on detecting viral DNA, irrespective of the possible infectivity of the virus. In contrast, flow cytometry assays, which identify viral surface proteins (i.e., GP64), may also detect viruses that are not actively replicating [63,106]. These limitations indicate that the infectivity of baculoviruses is typically evaluated using two conventional methods that rely on the cytopathic properties of the viruses: plaque assays and TCID50 (50% Tissue Culture Infectious Dose) assays [106]. Both approaches focus on the direct infectivity of the virus particles and the observation of changes in the insect cell line during infection. To estimate the virus titter, validation based on the lytic effect (in the plaque assay) and other cytopathic effects (in TCID50) necessitates about a week of monitoring for the infected cells (Figure 2). Moreover, accurately identifying microscopic changes related to the infection process (particularly in plaque assays) can be difficult for inexperienced researchers [107]. To lessen the labor-intensive nature of these assays, Hopkins and Esposito developed a new Sf9-Easy Titer (Sf9-ET) recombinant cell line [108]. They accomplished this by transfecting the Sf9 cell line in a stable manner with a plasmid-encoding eGFP under the control of the polh promoter, which is activated by baculovirus-specific transcription factors. Consequently, cells infected with the baculovirus produce a signal that can be easily detected using a fluorescence microscope (Figure 3). Observing fluorescence instead of cytopathic effects removed the challenge of assessing infection levels and reduced the time needed for virus titer determination to approximately three days [108]. Additionally, the insertion of the neomycin-resistant gene into the plasmid construct enabled the selection of monoclonal clones, which, after verifying the correct eGFP expression, were deposited as a stable transgenic cell line in the American Type Culture Collection. Since then, Sf9-ET has been routinely used for baculovirus titer determination during the preparation of infectious stocks [109,110].

Most recently, Kim et al. [111] reported the generation of a novel transgenic Sf9 cell line. The authors introduced an expression structure consisting of hr3 (homologous region 3), 39k, p10 promoters, and the eGFP-encoding sequence (hr3-p39k-pp10-eGFP) into the Sf9 genome using the piggyBac transposon system. After selecting transgenic Sf9 clones, they isolated the cell line with the highest and fastest fluorescence emission upon viral infection and named it Sf9-QE (Quick and Easy) (Figure 3) [111]. Their preliminary research suggests that this novel cell line outperforms the commercially available Sf9-ET in virus quantification due to its higher fluorescence expression over a shorter time. Although these properties of Sf9-QE were stable for at least 100 passages, further research is needed to fully evaluate the potential implementation of this clone in routine baculovirus titer determination [111].

### 4.4. The Implementation of the Adaptive Laboratory Evolution Approach in the Cultivation of Insect Cells

Adaptive laboratory evolution (ALE) has become a highly popular method for obtaining improved organisms in biomanufacturing. ALE approaches involve the long-term cultivation of organisms in precisely defined conditions with specific selection pressures (Figure 3). The ALE process produces newly evolved populations with enhanced phenotypes and increased fitness. While this method has been extensively applied to develop various improved strains of microorganisms for biomanufacturing purposes, the use of ALE in the context of cell lines, particularly insect cells, is just beginning to emerge with a primary focus on enhancing the yield of various VLPs.

The first application of ALE to insect cell lines was reported in 2014 [112]. To improve the yield of Chikungunya virus-like particles (VLPs), Wagner et al. [112] adapted the Sf21 insect cell line to a medium with a higher pH, characteristic of mammalian cultures, where the expression of these viral particles became detectable. The two-month adaptive cultivation with a progressively increasing medium pH led to the development of a new cell line, SfBasic, which demonstrated optimal growth within a pH range of 7.0–7.2 and produced Chikungunya VLP yields that were 11 times higher than those of the original Sf21 cell line, which grew in standard insect culture media with a pH of 6.2–6.4 [112]. The method of adapting insect cells for growth at higher pH levels was subsequently employed by Correia et al. [113], who achieved a three-fold increase in the production of influenza hemagglutinin (HA)-displaying VLPs using High Five™ cells that had been adapted to grow in a medium with a neutral pH compared to non-adapted cells.

Fernandes et al. [114] demonstrated the beneficial effect of another selection factor, i.e., temperature, on the expression of HIV-Gag VLPs. The authors showed that a three-month adaptation period for stable Gag-VLP-expressing cell lines (Sf9-Gag and Hi5-Gag) in hypothermic cultivation conditions (22 °C) increased the p24 protein yield (used as a proxy for HIV-Gag VLPs) by up to 26-fold in Sf9-Gag cells and 10-fold in Hi5-Gag cells compared to non-adapted cells cultivated for three months in standard (27 °C) conditions [114]. The ALE-obtained Sf9-Gag and Hi5-Gag cell lines were subsequently transfected with a hemagglutinin (HA)-encoding target cassette, resulting in the generation of new Sf9 Gag-HA and Hi5 Gag-HA insect cell clones capable of producing Gag-HA VLPs [115]. The productivity of the insect cell lines adapted to low temperatures was maintained upon pseudotyping with HA, showing a four-fold increase in HA and p24 yields in the temperature-adapted Sf9 Gag-HA cell line compared to the non-adapted clone. Moreover, the Sf9 Gag-HA cell line was adapted to operate in perfusion mode, enabling growth in high-cell-density cultures within a stirred-tank bioreactor and significantly enhancing process efficiency [115]. The combination of ALE and high-cell-density approaches facilitated the development of an efficient system for producing recombinant molecules, achieving HA titters approximately five times higher than those obtained with the standard insect cell BEVS approach [115]. These promising results were further optimized by defining the minimal cell-specific perfusion rate for the long-term continuous production of Gag-HA-pseudotyped VLPs at a 0.8 L working volume scale [116]. The pioneering work by Roldão’s team in implementing ALE to optimize stable insect cell lines as expression vectors for recombinant molecules underscores their potential as a viable alternative to BEVSs, even at an industrial scale (Figure 3) [117].

### 4.5. Glycoengineering of Insect Cell Lines

Jarvis’s research groups made pioneering advancements in glycoengineering insect cell lines. The first glycoengineered cell line, Sfβ4GalT, was developed in 1998. By transforming Sf9 cells with an expression plasmid encoding the sequence of bovine β1,4-galactosyltransferase, Hollister et al. [90] successfully isolated transgenic cells capable of adding β1,4-galactose to N-linked paucimannose glycans in a monoantennary manner. Following these findings, Hollister et al. [118] transformed the Sfβ4GalT cell line with α2,6-sialyltransferase genes, resulting in the first transgenic line that could produce monoantennary terminally sialylated N-glycans known as Sfβ4GalT/ST6. For the first time, biantennary glycoproteins were expressed in SfSWT-1 cells, which are transgenic clones of Sfβ4GalT that incorporate genes encoding five mammalian glycotransferases [119]. This demonstrated that insect cell lines can replicate human-like glycosylation patterns on recombinant proteins with minimal or no reduction in baculovirus infection rates [119]. In the ensuing years, Jarvis’s group enhanced the N-terminal sialylation of human glycans by further transforming the SfSWT-1 cell line with two additional mammalian genes encoding sialic acid synthase and CMP-sialic acid synthetase. This led to the creation of the SfSWT-3 cell line, which could produce N-terminally sialylated glycoproteins in a serum-free medium containing N-acetylmannosamine [120]. Significant advances followed the development of the first insect cell line, SfSWT-5, capable of activating the mammalian protein N-glycosylation pathway, differing from previous transgenic cell lines with “glycogenes” regulated by constitutive promoters [121]. In another study, Mabashi-Asazuma et al. [122] successfully integrated a human CMP-sialic acid transporter gene into the SfSWT-4 cell line, yielding stable SfSWT-6 cells with improved recombinant glycoprotein sialylation at lower concentrations of N-acetylmannosamine, thereby reducing the cost of manufacturing mammalian-like glycoproteins. Despite successfully developing insect cell lines capable of producing biantennary terminally sialylated mammalian N-glycans, the efficiency of the sialylation process was low, ranging from 0.1% (SfSWT-4) to 1% (SfSWT-6) of the total glycoproteins [122]. Swapping the promoter of the inserted glycogenes from the previously used constitutive ie1 promoter to the 39K promoter induced by baculovirus infection significantly increased N-glycan processing efficiency. The 39K promoter in the Sf39KSWT cell line supported higher glycogen expression levels, improving glycosyltransferase activity and N-glycan sialylation efficiency to 40% [123].

Advancements in genetic engineering that utilized the adaptive immune system of prokaryotes contributed to the development of the CRISPR (clustered regularly interspaced short palindromic repeats) Cas9 (*CRISPR*-*associated* protein *9*) tool, enabling efficient site-specific genome editing [124]. In nature, during prokaryotic infections, invader DNA is cut and incorporated into the CRISPR array as a new spacer sequence. The CRISPR array is then transcribed into pre-CRISPR RNA (pre-crRNA) and processed into mature CRISPR RNA (crRNA). These molecules assemble with one or more Cas proteins, forming a crRNA–Cas (crRNP) effector complex, which cleaves the complementary DNA sequence of reencountered, invasive DNA [124,125]. Researchers can mimic this process by designing guide RNA (gRNA) molecules as guides. These gRNAs consist of a crRNA, a 17–20 nucleotide sequence complementary to the target DNA, and trans-activating CRISPR RNA (tracrRNA) responsible for specific Cas9 binding and facilitating its enzymatic activity. With this technology, scientists can precisely target any sequence of interest and cut it with Cas9 endonuclease activity. The double-stranded breaks introduced within the sequence can be repaired by non-homologous end joining, typically leading to frameshift mutations and gene knockouts due to indel mutations at the joining site. Alternatively, double-stranded breaks can be rejoined by introducing an exogenous repair template, which can be incorporated into host DNA, allowing precise genome alterations [126,127].

The first successful application of CRISPR/Cas9 in insect genome editing was documented in 2013 by Gratz et al. [128]. The same year, Bassett et al. [129] induced deletions in the yellow and white genes of *Drosophila* embryos, resulting in mutagenesis of up to 88% of gRNAs injected into insects. Jarvis initiated the adaptation of CRISPR/Cas9 for protein glycoengineering, which is a still-ongoing process. Mabashi-Asazuma et al. [130] demonstrated that silencing the *FDL* gene in *Drosophila melanogaster*-derived S2+ cells reduces insect paucimannosidic structures in favor of mammalian-type N-glycans. In their study, the authors utilized CRISPR/Cas9 technology to introduce mutations at one of three selected targeting sites within the *FDL* gene sequence, which encodes the TM domain of fused lobe (FDL) proteins, referred to as DmFDLt1, DmFDLt2, and DmFDLt3. This resulted in the creation of three transgenic S2+-derived cell lines (DmFDLt1, DmFDLt2, and DmFDLt3) capable of attaching GlcNAc to the mannose precursor, thereby mimicking mammalian-like N-glycan patterns. The most efficient clone was the DmFDLt2 cells, which produced recombinant human erythropoietin, with 70% exhibiting a human-like glycosylation pattern [130]. The research conducted by Mabashi-Asazuma et al. documented the significant role of the FDL enzyme as a key factor distinguishing insect and mammalian N-glycan processing pathways. They highlighted that CRISPR/Cas9 could be used as an effective tool for editing insect cell genes, including those involved in glycoengineering. In subsequent years, Mabashi-Asazuma and Jarvis [130] adapted the CRISPR/Cas9 technique to efficiently introduce mutations within homologous sequences of the *FDL* gene in Sf9 and High Five™ cell lines. They accomplished this by constructing CRISPR/Cas9 vectors encoding gRNAs under homologous U6 promoters derived from *S. frugiperda* or *T. ni*, respectively, as the heterologous insect U6 promoters from *D. melanogaster* and *B. mori* were insufficient to drive gRNA expression in Sf9 or High Five™ cells. Introducing mutations in the SfFDLt1 region (analogous to DmFDLt1) using a CRISPR/Cas9 vector with the SfU6-3 promoter led to the development of the SfFDLt1 cell line, which could produce recombinant glycoproteins with approximately 90% of mammalian-like N-glycan elongation [131]. Most recently, Mabashi-Asazuma and Jarvis [132] announced the creation of the first insect cell line capable of producing glycoproteins with endo-β-N-acetylglucosaminidase H (Endo H)-cleavable N-glycans. They achieved this by inducing mutations in the *Mgat1* gene of the Sf-RVN cell line, which encoded the enzyme β1,2-N-acetylglucosaminyltransferase, which was responsible for forming Endo H-resistant N-glycan intermediates. The newly developed cell line, Sf-RVNLec1, is known for its efficient production of glycoproteins containing immature and high-mannose-type N-glycans that are sensitive to Endo H. Glycoproteins produced by this cell line, after the enzymatic cleavage of N-glycans, display enhanced crystallization, thus making them an excellent tool for researchers studying glycoprotein structures through X-ray crystallography. The studies conducted by Jarvis’s group and the improvement of N-glycan patterns in proteins derived from transgenic cell lines are illustrated in Figure 3.

Recently, efforts have also been made to adapt CRISPR/Cas9 technology to create cell lines with properties that improve the efficiency of recombinant protein production. Malmanche et al. [133] demonstrated the feasibility of knocking out the *Sf-caspase-1* gene in Sf9 cell lines using a CRISPR/Cas9 vector, successfully reducing caspase-1 expression and inducing resistance to apoptosis in the isolated clones. However, when the insect cells were infected with a recombinant baculovirus vector, there were no differences in the cell death rates or the amount of reporter protein produced between *naïve* and caspase-1 knockout cells [133]. Although the anticipated increase in recombinant protein production in cells lacking the active caspase-1 enzyme was not realized, this research highlights the potential application of CRISPR/Cas tools in cell engineering for biomanufacturing. Likewise, preliminary studies are being conducted on baculovirus vectors to improve the efficiency of recombinant protein production in insect cells [134,135].

## 5. The Application of BEVS Technology in the Production of Vaccines and Therapeutics

Since the first product was obtained using BEVS technology in 1983, i.e., heterologous human IFN-β [34], several licensed vaccines and therapeutics for humans and animals have become commercially available (Table 1).

The first commercially available veterinary subunit vaccine, Porcilis Pesti^®^, produced in insect cells, was developed for the classical swine fever virus [143] based on the E2 antigen and received European Market Authorization in 2000. Classical swine fever is classified as a notifiable disease by the WHO for animal health and is one of the most significant contagious diseases affecting pigs. The classical clinical form of this disease presents as an acute hemorrhagic condition with symptoms including high fever, depression, anorexia, and conjunctivitis. The disease has a very high morbidity and mortality rate, which can approach 100% [48,148].

Porcine circovirus type 2 (PCV2) belongs to the *Circoviridae* family and the *Circovirus* genus. It is the primary causative agent of several syndromes, collectively known as porcine circovirus-associated disease (PCVAD). PCVAD is a rapidly emerging global issue that significantly affects swine-producing nations and is arguably one of the most economically significant diseases impacting the global swine industry. Furthermore, many syndromes associated with PCVAD arise from coinfection with PCV2 and other pathogens, such as *Mycoplasma* and *Lawsonia* [149,150]. Currently, five subunit vaccines are licensed for use in the international market, all of which utilize the expression of the ORF2 protein in a baculovirus system. Three are monovalent vaccines (PCV2), while two include inactivated *Mycoplasma hyopneumoniae* and *Lawsonia intracellularis* (Table 1).

The regulatory acceptance of BEVSs in manufacturing human vaccines marked a significant milestone with the approval of Cervarix™, the first BEVS-derived vaccine against cervical cancer, in 2007 [151]. Cervarix is a bivalent vaccine that comprises VLPs made of human papillomavirus L1 capsid proteins targeting human papillomavirus types 16 and 18. In 2013, FluBlok became the first recombinant BEVS-derived protein vaccine approved to prevent seasonal influenza. FluBlok and FluBlok Quadrivalent contain three or four types of recombinant hemagglutinin antigens derived from influenza A and B viruses [38].

On 10 December 2021, following the recommendation of the European Medicines Agency, the European Commission granted conditional marketing authorization for the BEVS-derived subunit vaccine against SARS-CoV-2, branded Nuvaxovid [137]. It is the only SARS-CoV-2 vaccine that employs a more traditional production technique for the spike protein, making it suitable for individuals cautious about new-generation mRNA technology. The manufacturer labels Nuvaxovid as a nanoparticle vaccine, although it can also be classified as a subunit vaccine. It consists of a modified spike protein with prolines replacing the original residues (K986P and V987P) to maintain the pre-fusion structure of the glycoprotein and enhance resistance to proteolytic cleavage. The protein is produced in the Sf9 cell line, integrated into cell membranes, harvested, and assembled into lipid nanoparticles, using the M-Matrix as an adjuvant [152] (Figure 4).

Provenge (Sipuleucel-T, Dendreon) was the first therapeutic cancer vaccine approved by the U.S. Food and Drug Administration in April 2010. It is an immunotherapy product comprising autologous dendritic cells loaded ex vivo with a recombinant fusion protein that combines prostatic acid phosphatase and the granulocyte-macrophage colony-stimulating factor. The vaccine is intended for patients with asymptomatic or minimally symptomatic metastatic castration-resistant prostate cancer [138,139].

The genetic therapy Glybera was developed by uniQure in Amsterdam and received five-year approval from the European Union in 2012. It utilizes insect cells and recombinant baculovirus technology. The drug has not been registered in the United States. Glybera was an adeno-associated viral vector engineered to express lipoprotein lipase in muscle tissue and to treat rare congenital lipoprotein lipase deficiency. A single dose costs EUR 1 million and was administered to only one patient. The company opted not to pursue re-registration, and Glybera remained available in the EU until 25 October 2017 [48,141,154].

In 2008, Hinke [155] created a BEVS that employed a recombinant 65 kDa glutamate decarboxylase, which corresponds to human glutamate decarboxylase 65, which is the main [MS1] autoantigen involved in type 1 diabetes. Recently, in July 2024, Diamyd^®^ (rhGAD65/alum) received the second Fast Track designation from the U.S. Food and Drug Administration to treat type 1 diabetes in pediatric patients at stage 1 or stage 2 with the HLA DR3-DQ2 genotype. Earlier this year, Diamyd^®^ also gained Fast Track designation for treating individuals with stage 3 type 1 diabetes who possess the HLA DR3-DQ2 genotype [142].

With increasing BEVS derivatives gaining approval from global regulators and advancing into clinical phases, the level of public knowledge of BEVSs is steadily rising. After 40 years of advancement, relevant regulations and laws have significantly improved, encouraging academic researchers to investigate BEVSs for vaccine development. Public awareness of BEVS vaccines has also expanded. Today, BEVSs are one of the leading technologies in vaccine production.

In addition to commercially available SARS-CoV-2 and FluBlok vaccines, many BEVS-derived vaccines are or have been in the preclinical and clinical trial phases (Table 2).

## 6. Pros and Cons of BEVSs in the Production of Recombinant Proteins and Vaccines

BEVSs provide numerous advantages, including rapid production, flexible product design, improved safety, and scalability. These features position BEVSs as leading platforms for developing recombinant subunit vaccines, although they may not be suitable for every product. The exceptional safety profile of BEVSs is especially appealing to regulatory agencies. Baculoviruses specifically target insect cells, and it was confirmed by the European Commission’s Health and Consumer Protection Directorate-General in 2008 that they do not pose any risks to human health and do not lead to pathogenic, carcinogenic, or genotoxic effects in mammalian cells [38]. Without selection pressure, the baculovirus genome neither integrates into the host cell genome nor replicates within it [177].

Post-translational modifications, including phosphorylation, glycosylation, ubiquitination, and acetylation, are crucial for producing active recombinant proteins. BEVS platforms mainly facilitate these modifications [178]. The capacity of the baculovirus genome to incorporate multiple genes and support post-translational modifications makes the BEVS platform especially attractive for expressing complex or challenging proteins, such as various enzymes, parasitic proteins, glycoproteins, and VLPs [48,179,180]. Moreover, it benefits functional studies, crystallography, and drug discovery research [181,182].

Compared to traditional vaccine production methods, BEVS platforms enable faster production and greater scalability. For instance, developing a seasonal influenza vaccine using chicken embryos usually takes about six months, which may not align with the circulating strain of the influenza virus [183]. In contrast, BEVSs reduce the vaccine preparation time to just one and a half months, allowing for a quicker response to emerging influenza outbreaks [184]. Furthermore, BEVS technology eliminates the costs associated with pathogen isolation since it does not require the handling of live pathogens. As a result, products created through the recombinant BEVS process are free from pathogens, eggs, and the most potentially harmful or allergenic chemicals, ensuring their purity [48].

The reduced production costs associated with BEVS-based preparations are noteworthy, highlighting the economic viability of this method, which, in turn, leads to lower prices for consumers. This is attributed mainly to insect cells that develop in a cost-effective, serum-free medium at high densities [185]. Research shows that the efficiency of recombinant protein production in SF+ cells can range from 2 to 21,000 L and is achievable commercially in bioreactors of various sizes [48].

However, BEVSs have specific limitations, particularly concerning post-translational glycosylation, which does not match the complexity found in higher eukaryotes. Therefore, producing proteins that require complex post-translational modifications and assembly is more effectively accomplished through mammalian expression systems [120,186,187].

Another drawback of the BEVS platform is its potential to trigger an inflammatory response that is characterized by the production of inflammatory cytokines and the activation of both the classical and alternative complement pathways [188]. This response can lead to viral genome degradation, reduced transgene expression, unstable protein production, and the instability of the baculovirus genome [188]. Nevertheless, recent vector optimization and genome modification advancements have significantly improved the BEVSs [189]. Moreover, the activation of the classical and alternative complement pathways can be mitigated by incorporating the decay-accelerating factor H-like protein-1, C4b-binding protein, and membrane cofactor protein onto the baculovirus envelope [190,191]. In conclusion, the capacity for large-scale production, the advantages of proper protein folding and post-translational modifications, and the rapid production timelines make the BEVS platform an attractive option for future biological applications [48]. The potential of utilizing insect cell lines in recombinant protein manufacturing is evident in market analysis. Currently valued at approximately USD 2.5–3.0 billion, the global recombinant protein market attributes up to 10% of its revenue to products derived from insect cell lines [192,193]. The market is projected to grow at a compound annual growth rate (CAGR) of around 10.2% between 2025 and 2030, reaching an estimated value of USD 5.58 billion, with the share of insect cell lines expected to remain consistent [192]. A forecast analysis focusing on the total value of the insect cell lines market predicts growth at a CAGR of 12% during the 2024–2036 period, primarily highlighting the biopharmaceutical manufacturing segment, particularly the involvement of cell lines in the production of vaccines and recombinant proteins [193]. These trends underscore the current use of insect cell lines and signal long-term investments to develop innovative approaches that enhance their efficiency in biopharmaceutical manufacturing.

## 7. Limitations and Challenges in Implementing Insect Expression Systems for Biopharmaceutical Production

Although market trends indicate a growing potential for insect cell lines in the biopharmaceutical sector, limitations must be addressed to fully harness their potential in developing new therapeutics. In systems based on bacmid technology, a particular concern is the presence of bacterial sequences that may contaminate the final product. DNA fragments such as antibiotic-resistant genes can be encapsulated in recombinant AAVs or VLPs and mistakenly delivered to patients [194,195,196]. This raises significant safety concerns for regulatory agencies like the EMA and FDA. Another issue relates to the consistency of the biopharmaceutical production process associated with bacmid-based technologies. After multiple passages (>5), bacmid-derived baculoviruses tend to lose genomic stability, ultimately resulting in the loss of GOI expression and a dramatic decrease in the recombinant protein yield. Although studies in this area are scarce, the root of the problem is believed to lie in the presence of the bacteria-origin mini-F replicon found in the bacmid backbone, located near the GOI inserted within the Tn7 transposition site. This “foreign” element, non-native to insect viruses, appears to be a significant target of selection pressure during baculovirus replication in insect cells. This pressure promotes multiple deletions within the mini-F sequence, which also compromises the stability of the GOI [197,198]. To reduce the presence of bacterial-origin sequences that impact safety and reproducibility, non-bacmid linearized baculovirus systems based on homologous recombination in insect cells have been introduced (e.g., flashBAC™, FlexiBac) [109,199]. A novel approach for minimizing unwanted sequences involves synthetic biology, which enables the construction of functional viral particles from chemically synthesized DNA fragments [200]. This allows for the precise manipulation of the baculovirus genome at the single-nucleotide level, which will undoubtedly play a key role in future biopharmaceutical developments.

An alternative to baculovirus-free methods involves using genetically transformed insect cell lines that enable constitutive protein expression. While this approach is gaining increasing recognition for recombinant expression at the laboratory scale, it remains far from being implemented on an industrial scale in the biopharmaceutical sector. The challenge lies in the expression efficiency of constitutive promoters and the significant costs associated with working with transformants, which require continued antibiotic selection to maintain genomic stability and consistent protein expression [195,201]. These costs become substantial at bioreactor scales, especially since the commonly used selective antibiotics for insect cell lines, such as Zeocin™, Blasticidin™, or Geneticin™, are relatively expensive. Although promising, various approaches aimed at enhancing protein yield and production capacity through implementing modified insect cell lines are still limited to laboratory-scale applications. Scaling up selected clones for biopharmaceutical purposes requires cell banking according to Good Manufacturing Practice (GMP) guidelines. The production and maintenance of GMP-grade materials involve significant expenditures to ensure safety, quality, and reproducibility. Moreover, the use of such reagents is mandatory in the manufacture of medical-grade biopharmaceuticals [202]. These factors contribute to increased maintenance costs and complexity in large-scale insect cell line production.

Regardless of whether recombinant protein production is conducted using baculovirus-based or virus-free methods, there remains significant concern regarding the contamination of insect cells with adventitious viruses. This issue has been documented for all routinely used insect cell lines for the recombinant protein [203]. It also affects the most commonly used lines derived from *S. frugiperda* (Sf9, Sf21) and *T. ni* (High Five™), which are contaminated with rhabdoviruses (Sf-rhabdovirus) and alphanodaviruses (Trichoplusia ni cell line virus), respectively [203]. Since these viruses do not cause apparent cytopathic effects, their presence went unnoticed for decades. They were discovered during the analysis of generated virus-like particles, which revealed the presence of foreign genetic sequences unrelated to the production target, such as rhabdoviridae sequences [204] or even alphanodavirus-like particles [205]. Although studies conducted so far have not demonstrated the pathogenicity of these viruses to humans, they raise significant safety concerns, particularly the Sf-rhabdovirus, which belongs to the same family as the hazardous rabies virus. Regulatory agencies mandate extensive virus testing and clearance procedures during biopharmaceutical production; therefore, insect cells must be free of such agents. Consequently, the biopharmaceutical industry currently employs insect cell lines that are free from adventitious viruses, the maintenance and monitoring of which incur additional costs [206].

Although significant progress has been made, it is worth noting that the biopharmaceutical industry still faces significant challenges in translating lab-scale experiments into industrial-scale production, especially regarding the integration of enhanced insect cell lines that comply with strict regulatory standards. This process remains both technically complex and costly.

## 8. Conclusions

Although more than a hundred years have passed since the advent of insect cell cultures, the last thirty years have been groundbreaking. The development of bacmid technology, which differentiates insect cell lines from other protein expression vectors, has paved the way for various vaccines and therapeutic preparations. An intriguing alternative to baculovirus-based systems is using stable cell lines that produce proteins and recombinant protein complexes, including VLPs. In this context, the recently adopted ALE approach is especially noteworthy. Early studies indicate that this method can enhance the efficiency of recombinant protein production in stable cell clones. Using similarly efficient baculovirus-free methods provides a viable alternative to BEVSs, offering greater flexibility and simplifying final product purification. Another emerging system uses insect larvae as vectors for recombinant protein production. Although the reports on this approach are still limited, the research findings are promising, and the production costs seem lower than those associated with insect cell lines. In the future, introducing the CRISPR/Cas9 technology for germline genome modification could pave the way for establishing highly efficient recombinant protein factories based on larvae. Like stable cell lines, this approach could eliminate the need for baculoviruses. However, insect cell lines remain a crucial component of recombinant biomanufacturing, as evidenced by Nuvaxovid^®^, the only approved subunit vaccine against SARS-CoV-2 in Europe, which continues to be updated.

## Figures and Tables

**Figure 1 vaccines-13-00556-f001:**
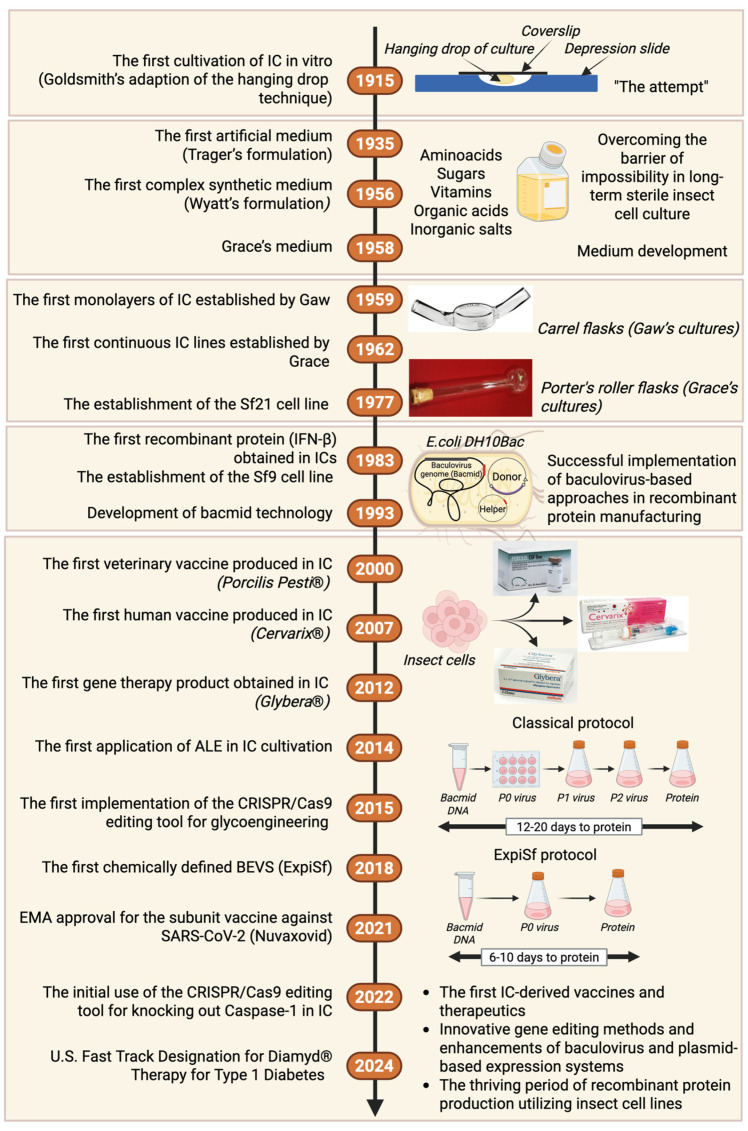
Milestones in developing insect cell cultures. IC—insect cells; ALE—adaptive laboratory evolution; BEVS—baculovirus expression vector system. Sources of pictures: [21,22,23,24,25]. ExpiSf vs. the classical protocol for protein production in insect cells adapted from [26]. Created with BioRender.com (accessed on 20 January 2025).

**Figure 2 vaccines-13-00556-f002:**
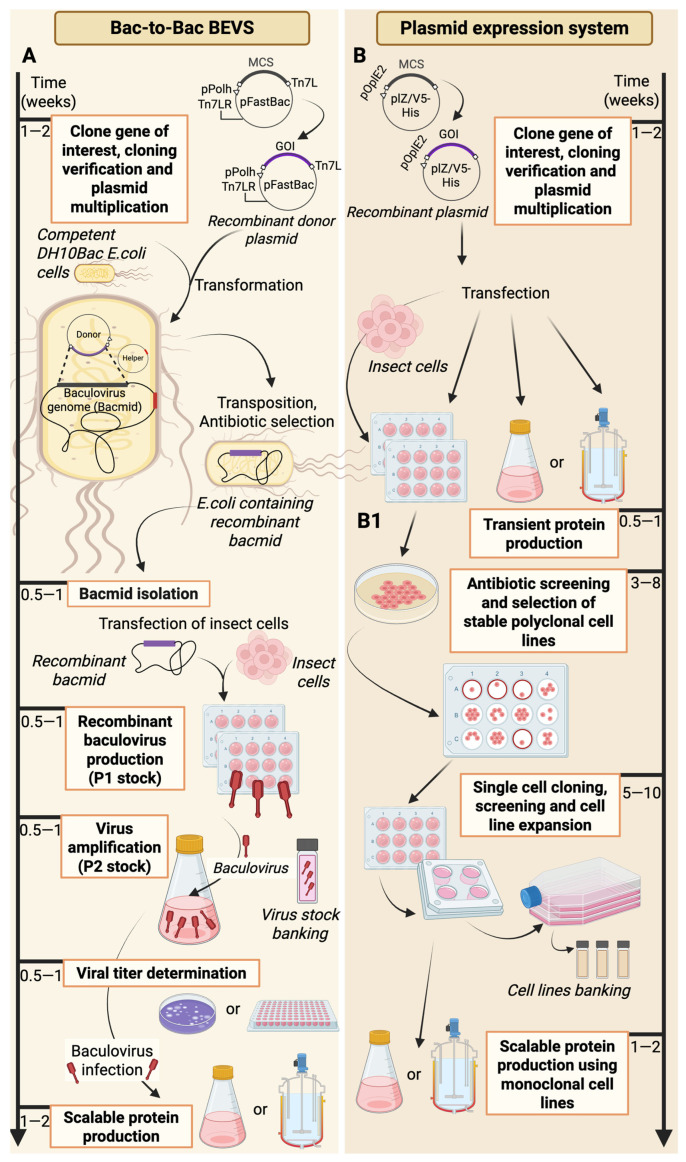
Recombinant protein production using BEVSs, emphasizing the classical Bac-to-Bac system (**A**) and baculovirus-independent, plasmid-based transient transfection (**B**). For (**B**), stable monoclonal cell lines can be established, characterized by the constitutive production of recombinant proteins (**B1**). The duration (in weeks) indicated for each stage is an approximation. It can be gradually reduced, for example, by implementing new BEVSs, such as ExpiSf (refer to Figure 1 for a comparison of the systems) or by replacing traditional virus titer determination methods based on cytotoxic effects with FACS or qPCR methods (in scenario **A**). The time needed to obtain monoclonal cell lines can also be extended (in scenario **B1**) when an additional clone selection is carried out to achieve the highest recombinant production, alongside stability testing of the clones through increased numbers of passages. Created with BioRender.com (accessed on 21 January 2025).

**Figure 3 vaccines-13-00556-f003:**
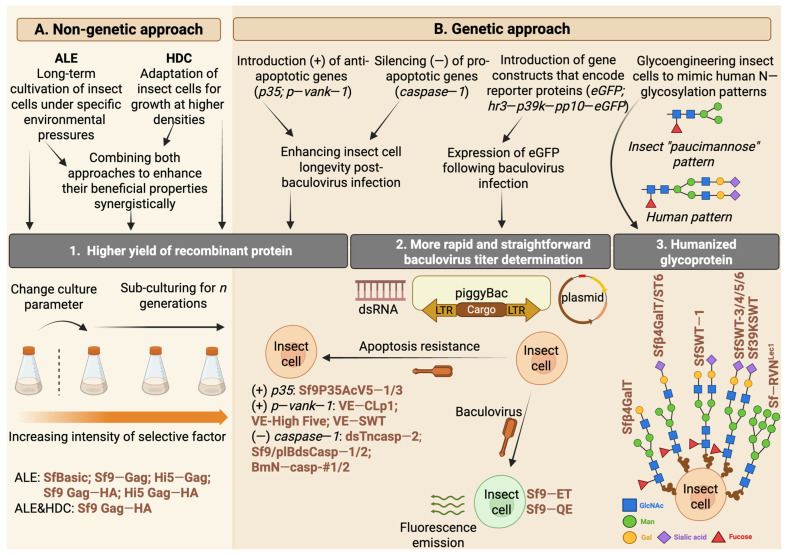
An overview of different strategies to increase the potential of insect cell lines in the production of recombinant proteins. Strategies are categorized into those based on non-deliberate genetic approaches (**A**) and those involving the targeted genetic engineering of insect cell lines (**B**). Strategies (**A**), which include adaptive laboratory evolution (ALE) and high-density culture approaches (HDC), contribute to an increase in the yield of recombinant products. Strategies (**B**) have been successfully applied to (1) increase the recombinant product yield, (2) facilitate the determination of baculovirus titers, and (3) achieve humanized N-glycosylation patterns in recombinant proteins. The names of the novel stable cell lines derived from the aforementioned studies are shown in brown. The glycosylation patterns next to the cell line names indicate the progress of specific clones in developing human-like glycosylation patterns. In the case of the Sf-RVNLec1 cell line, the N-glycosylation obtained with an Endo H-sensitive motif enables the study of glycoprotein structures. GlcNac—N-acetylglucosamine; Man—mannose; Gal—galactose. Created with BioRender.com (accessed on 21 January 2025).

**Figure 4 vaccines-13-00556-f004:**
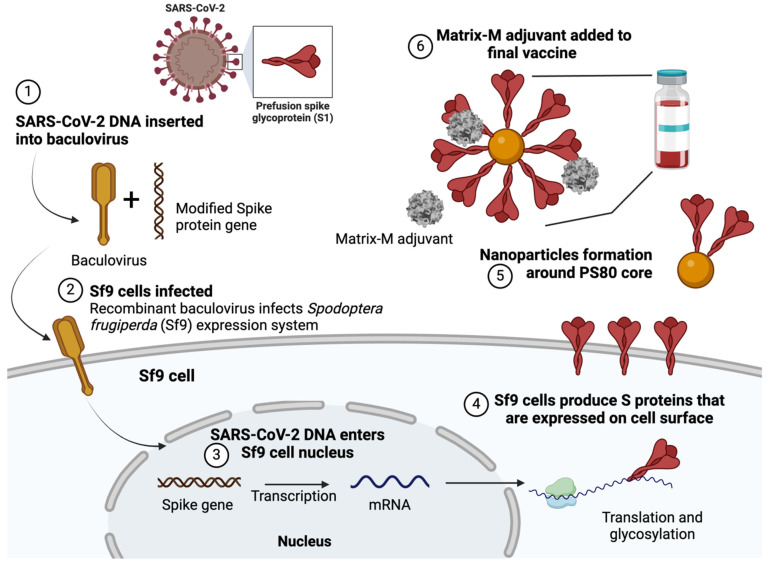
A diagram illustrating the Nuvaxovid vaccine manufacturing process using a baculovirus and *Spodoptera frugiperda* cell line platform. Created with BioRender.com (accessed on 19 January 2025). Adapted from NVX-CoV2373-Vaccine-Design, p. 6 [153].

**Table 1 vaccines-13-00556-t001:** Vaccines and therapeutics for humans and animals derived from BEVSs that are currently or have been commercially available.

Disease	Brand Name	Manufacturer	Antigen/Product Type	References
**Human Vaccines**				
Cervical cancer	Cervarix^TM^	GSK	L1 capsid protein/VLP	[136]
Influenza	FluBlok^®^	Protein Sciences Corporation	Hemagglutinin/subunit	[48]
Influenza	Flublok^®^ Quadrivalent			[48]
SARS-CoV-2	Nuvaxovid^®^/Covovax	Novavax	S protein/subunit	[137]
**Human therapeutics**				
Prostate cancer	Provenge^®^	Dendreon	PSA/immunotherapy	[138,139,140]
Lipoprotein lipase deficiency	Glybera^®^	uniQure	rAAV-based gene therapy	[48,141]
Type 1 diabetes	Diamyd^®^	Diamyd Medical	65 kDa glutamate decarboxylate	[142]
**Veterinary vaccines**				
Porcine circovirus type 2	Porcilis^®^ PCV	Schering-Plough/Merck	PCV2 ORF2 protein/VLP	[143,144]
Porcine circovirus type 2	Ingelvac CircoFLEX^TM^	Boehringer Ingelheim	PCV2 ORF2 protein/VLP	[143,145]
Porcine circovirus type 2	Circumvent^®^	Intervet/Merck	PCV2 ORF2 protein/VLP	[143]
Porcine circovirus type 2 and *Mycoplasma hyopneumoniae*	Porcilis PCV M Hyo^®^	Intervet/Merck	PCV2 ORF2 protein/VLP and inactivated *Mycoplasma hyopneumoniae*	[144]
Porcine circovirus type 2, Mycoplasma hyopneumoniae and *Lawsonia intracellularis*	Circumvent CML^®^	Merck	PCV2 ORF2 protein/VLP, inactivated *M. hyopneumoniae* and *L. intracellularis*	[146]
Classical swine fever	Porcilis Pesti^®^	Boehringer Ingelheim/Merc	E2 protein/subunit	[147]
Classical swine fever	Bayovac CSF E2^®^	Bayer/Pfizer	E2 protein/subunit	[48]

**Table 2 vaccines-13-00556-t002:** BEVS-derived clinical and preclinical vaccines.

Target	Antigen	Manufacturer	Stage/NCT Number	References
SARS-CoV-2	Recombinant RBD monomer	West China Hospital of Sichuan University	Phase III (NCT04904471)	[156]
SARS-CoV-2	CoV-2 preS dTM	Sanofi/GSK	Phase III (NCT04904549)	[157]
SARS-CoV-2	SARS-CoV-2 spike glycoproteins	Novavax	Phase I/II (NCT04368988)	[158]
SARS-CoV-2	SARS-CoV-2 spike glycoproteins	Radboud University Medical Center (Netherlands)	Phase I (NCT04839146)	[159]
Influenza A H1N1	A (H1N1) 2009 Influenza VLP	Novavax	Phase II (NCT01072799)	[160]
Seasonal Influenza virus	Hemagglutinin, neuraminidase, and Matrix-M adjuvant	Novavax	Phase III (NCT04120194)	[161]
SARS-CoV-2 and Influenza	Quadrivalent Influenza hemagglutinin and CoV-2 rS of SARS-CoV-2 nanoparticles	Novavax	Phase III (NCT06291857)	[162]
Human parvovirus B19	VP1 and VP2	National Institute of Allergy and Infectious Diseases/Meridian Life Science	Phase I/II (NCT00379938)	[163]
Respiratory Syncytial Virus	Fusion glycoprotein	Novavax	Phase III (NCT02624947)	[164]
Norwalk virus	Norwalk virus-VLP	Baylor College of Medicine	Phase I/II (NCT00973284)	[165]
Norwalk virus	Norwalk virus-VLP	LigoCyte Pharmaceuticals	Phase I/II (NCT00806962)	[166]
Malaria	ChAd63-MVA ME-TRAP	Novavax	Phase I/IIb (NCT01635647)	[167]
Ebola virus (EBOV)	EBOV surface glycoprotein	Novavax	Phase I (NCT02370589)	[168]
Human papillomavirus	HPV (6/11/16/18/31/33/ 35/39/45/51/52/56/ 58/59) L1 protein	SinoCellTech (China)	Phase III (NCT06041061)	[169]
Influenza A H7N9	Hemagglutinin	Animal Infectious Disease Laboratory, School of Veterinary Medicine, Yangzhou University	preclinical	[170]
Marburg virus (MARV)	MARV surface glycoprotein	Virology Division, United States Army Medical Research Institute for Infectious Diseases	preclinical	[168,171]
Zika virus (ZIKV)	ZIKV E protein	College of Veterinary Medicine and Institute of Veterinary Science, Kangwon National University, Korea	preclinical	[172]
Zika virus (ZIKV)	ZIKV prM/E	Department of Bioindustrial Technologies, Konkuk University, Seoul, Korea.	preclinical	[173]
Yellow fever virus (YFV)	YFV E-NS1 protein	Laboratorio de Patogénesis Viral, Instituto de Biotecnología y Biología Molecular (IBBM), CONICET-UNLP, La Plata, Buenos Aires, Argentina	preclinical	[174]
West Nile virus (WNV)	preM/E proteins	State Key Laboratory of Agricultural Microbiology, Huazhong Agricultural University, Wuhan, Hubei, China	preclinical	[175]
Dengue virus (DENV)	DENV E protein	Viral Disease and Vaccine Translational Research Unit, Institute Pasteur of Shanghai, China	preclinical	[176]

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
