# Peer review of "The Bioengineering of Insect Cell Lines for Biotherapeutics and Vaccine Production: An Updated Review"

_vaccines, 2025, doi:10.3390/vaccines13060556_

Round 1
Reviewer 1 Report
Comments and Suggestions for Authors
The review titled “An Updated Review of Improvements and Application of Insect Cell Lines in the Production of Vaccines and Therapeutics” shows a deep summary about the bioengineering applied to the use of insect cell lines in the production of recombinant proteins from different origins, focused on yield increase and glycoengineering of the human recombinant proteins with therapeutical application; the use and evolution of baculovirus and plasmid-based expression vectors specific to insect cell lines is explained. Also, the current use of these powerful tools to develop and to evaluate vaccines is discussed.
The current version of the manuscript is so interesting and detailed, I have comments that can help to improve it:
Major comments
-I suggest a change in the titled of the review to: Bioengineering of Insect Cell Lines for the Biotherapeutics and Vaccines production: an updated review.
-The development and management of the Sf insect cell lines can be better understood in a timeline scheme. Please, include it.
-Change the titled of the section 6 to “Pros and Cons of BEVSs in the production of recombinant proteins and vaccines”
-Although is explained briefly in the conclusions, please include a section of the future about these tools and other applications and uses.
Minor comments
In the Line 199 says “necessitating”, it could be better “which require or needing”
In the line 270 says “circumventing”, it could be better “avoiding”
Comments on the Quality of English LanguageThe manuscript needs a proofreading English writing.
Author Response
The review titled “An Updated Review of Improvements and Application of Insect Cell Lines in the Production of Vaccines and Therapeutics” shows a deep summary about the bioengineering applied to the use of insect cell lines in the production of recombinant proteins from different origins, focused on yield increase and glycoengineering of the human recombinant proteins with therapeutical application; the use and evolution of baculovirus and plasmid-based expression vectors specific to insect cell lines is explained. Also, the current use of these powerful tools to develop and to evaluate vaccines is discussed.
The current version of the manuscript is so interesting and detailed, I have comments that can help to improve it:
Authors: We sincerely thank you for your kind words regarding our manuscript and for the valuable suggestions that have helped improve its quality. We have made every effort to address and incorporate your comments. We hope that the changes we introduced meet your expectations.
Reviewer: I suggest a change in the titled of the review to: Bioengineering of Insect Cell Lines for the Biotherapeutics and Vaccines production: an updated review.
Authors: We have updated the title based on the Reviewer's suggestions.
Reviewer: The development and management of the Sf insect cell lines can be better understood in a timeline scheme. Please, include it.
Authors: In the timeline figure (Line 119), we have added information regarding the year the Sf9 cell line was established. This complements the existing details in the timeline about the parental Sf21 line and the ExpiSf system, which is based on a derivative of Sf9 cells.
We are concerned that further expansion might compromise the figure's readability, given the amount of information already presented. We hope this additional detail will be sufficient to maintain its clarity.
Reviewer: Change the titled of the section 6 to “Pros and Cons of BEVSs in the production of recombinant proteins and vaccines”
Authors: We changed the title of section 6 as suggested by the Reviewer.
Reviewer: Although is explained briefly in the conclusions, please include a section of the future about these tools and other applications and uses.
Authors: Thank you very much for your suggestion. Considering your suggestion and that of another reviewer, we decided to introduce a section dedicated to the limitations and challenges associated with the use of cell lines in the pharmaceutical industry (Lines 776-842). This issue currently requires researchers' attention regarding the scaling up from the laboratory to the industrial level and the future implementation of the described new tools and advances in the biopharmaceutical industry.
Reviewer: In the Line 199 says “necessitating”, it could be better “which require or needing”. In the line 270 says “circumventing”, it could be better “avoiding”
Authors: We introduced all these suggested changes.
Reviewer: The manuscript needs a proofreading English writing.
Authors: An English speaker checked the text, and all suggested changes have been implemented.
Reviewer 2 Report
Comments and Suggestions for Authors
This is a very comprehensive review, and the authors address a very interesting topic.
These are only minor details that I suggest you take into account to comply with the journal's format.When authors include more than one citation, only one is relevant.
Line 91 “the artificial media he created [13, 14]”, it seems to me that only [14] should be considered
Line 130 “studies of insect physiology and served as virus vectors for exploring animal and plant diseases [27–30]”, it seems to me that only [27] should be considered
Line 310 “this capability has led to the system being widely adopted to produce various recombinant molecules [61, 66–68]” Consider only the relevant reference
Example of correct citation Ross G. Harrison [6].
Cite correctly (examples)
Line 101 Gaw et al.
Line 131 Vail et al.
Line 156 Smith et al.
Line 196 Calles et al.,
Line 394 Cartier et al.,
Line 396 Lin et al.,
Line 415 March et al.
Line 396 Fath-Goodin et al
Author Response
Reviewer: This is a very comprehensive review, and the authors address a very interesting topic.
Authors: Thank you very much for your kind words about our work and for the valuable editorial suggestions, which we have made every effort to address.
Reviewer: These are only minor details that I suggest you take into account to comply with the journal's format. When authors include more than one citation, only one is relevant.
Line 91 “the artificial media he created [13, 14]”, it seems to me that only [14] should be considered
Line 130 “studies of insect physiology and served as virus vectors for exploring animal and plant diseases [27–30]”, it seems to me that only [27] should be considered
Authors: We introduced all these suggested changes.
Reviewer: Line 310 “this capability has led to the system being widely adopted to produce various recombinant molecules [61, 66–68]” Consider only the relevant reference
Authors: In this comment, we would like to retain the indicated references. This is because the paragraph aims to show the reader that the described ExpiSf system has been successfully adapted for the production of various recombinant molecules. To avoid unsupported claims, we refer to the production of SARS-CoV-2 VLPs (Mi et al.), PCV2b- and PCV2d-based virus-like particles (Kim & Hahn), AAV vectors (Kurasawa et al.), and membrane proteins (Kaipa et al.). We believe that presenting a range of diverse molecules supports the scientific validity of our paragraph.
Reviewer: Example of correct citation Ross G. Harrison [6].
Cite correctly (examples)
Line 101 Gaw et al.
Line 131 Vail et al.
Line 156 Smith et al.
Line 196 Calles et al.,
Line 394 Cartier et al.,
Line 396 Lin et al.,
Line 415 March et al.
Authors: Thank you for bringing this to our attention. We have corrected all of the examples mentioned, as well as other similar instances found in lines 147, 176, 331, 356, 360, 399, 463, 485, 493, 496, 537, 539, 554, 584, 597, 601, 610 and 624
Reviewer 3 Report
Comments and Suggestions for Authors
Authors should use the current literature on the subject. Use the literature of the last 30 years in the article.
Author Response
Authors should use the current literature on the subject. Use the literature of the last 30 years in the article.
Authors: Thank you for your thoughtful reading of the manuscript and your insightful suggestions.
Regarding the recommendation to limit citations older than 30 years, we respectfully clarify that several references were intentionally retained because they cite original, pioneering studies. These works provide essential historical context that is crucial for understanding the development of insect cell lines and the early applications of baculovirus systems. Omitting these original references would overlook key milestones in the evolution of the field and, in our view, would represent a significant oversight. Nevertheless, we have reduced the number of such references and removed the papers by Hirumi & Maramorosch (1964), Mitsuhashi & Maramorosch (1964), and Tokumitsu & Maramorosch (1966) to address your concern. At the same time, to ensure the manuscript better reflects current developments, a new paragraph has been added discussing the limitations and future perspectives of insect cell line use in the biopharmaceutical sector. Most of the literature cited in this section is less than 10 years old. Thank you for your understanding.
Reviewer 4 Report
Comments and Suggestions for Authors
This is a wide ranging review of insect cell-based expression systems for recombinant protein production. The authors trace the development of these systems right back to the very early days of culturing insect cells, before the use of antibiotics or defined growth media. The review then describes the early development of the baculovirus expression system and continues with an update on its current status. Historically, the exploitation of baculoviruses has been a slow burn affair, taking many years to become widely accepted as a system for the production of therapeutics, including vaccines.
Overall, I have few criticism of the review. Such a body of work is open to interpretation in various ways but the authors have been quite even-handed in their assessment of the technology. I offer the following comments as a second opinion on the field. The authors are free to take note of these or ignore them as they choose.
- Baculovirus systems for the production of therapeutic proteins.
The authors focus heavily on the use of Bac-to-Bac as a vehicle for protein production. However, I don't know of any commercial product that uses this system. B2B is a great research tool but suffers two major disadvanteages. The first is that bacterial sequences are retained within the virus vector. The authors do acknowledge this but in the same breath almost seem to ignore it. Regulators do not like these sequences in any production system. The second problem is one that is difficult to address as I am not aware of any publication that reports on it. This is the fact that B2B recombinants are inherently unstable for long term protein production. Once you passage virus past 4 growth cycles in cell culture the yield of protein production falls dramatically. By P8 it is non-existent. This is likely due to the bacterial sequences retain within the virus genome, although nobody has addressed this experimentally. At scientific meetings the phenomenon has been described but there is nothing in the literature to my knowledge. Therefore, all current commercial products are made using bacterial sequence-free vectors. - Continous protein production in insect cells.
While the use of stably transformed insect cells is attractive for recombinant protein production, the generally much lower yields make this economically difficult. There is also a problem of genetic stability of these cells as without continued antibiotic (e.g. G418) selection) the level of transgene expression declines.
The cited advantage of reduced cell debris in the biomass prior to purification is also over emphasized. If baculovirus-infected cells are harvested in a timely manner then cell culture medium contains few of these contaminants. - Modified insect cells for enhanced protein production with baculoviruses.
There have been some very clever works performed to modify insect cells for improved recombinant protein production. Although some of these are hard to reproduce from personal experience. While fine in principle, for future commercial production of clinical products they will need to be banked as GMP reagents. Both master and working cell banks have to be made. This is a very expensive process. - Virus contaminants in insect cells.
The authors have completely ignored the nodavirus contamination of Hi5 cells and the rhabdovirus presence in Sf cells. Any cell line used for clinical production of proteins must now be free of these agents. This is the one point that I would insist the authors include in their review.
We await with some tension before the FDA find other viruses to be removed from insect cells!
So in summary, I am much in favour of this review. It is broad in its scope and detail. It is an enjoyable read and will prove invaluable for those new to the field and for those with experience who just want to get up to date with current developments.
Finally, correct the spelling of "verification" in Fig.2.
Author Response
This is a wide ranging review of insect cell-based expression systems for recombinant protein production. The authors trace the development of these systems right back to the very early days of culturing insect cells, before the use of antibiotics or defined growth media. The review then describes the early development of the baculovirus expression system and continues with an update on its current status. Historically, the exploitation of baculoviruses has been a slow burn affair, taking many years to become widely accepted as a system for the production of therapeutics, including vaccines.
Overall, I have few criticism of the review. Such a body of work is open to interpretation in various ways but the authors have been quite even-handed in their assessment of the technology. I offer the following comments as a second opinion on the field. The authors are free to take note of these or ignore them as they choose.
- Baculovirus systems for the production of therapeutic proteins.
The authors focus heavily on the use of Bac-to-Bac as a vehicle for protein production. However, I don't know of any commercial product that uses this system. B2B is a great research tool but suffers two major disadvanteages. The first is that bacterial sequences are retained within the virus vector. The authors do acknowledge this but in the same breath almost seem to ignore it. Regulators do not like these sequences in any production system. The second problem is one that is difficult to address as I am not aware of any publication that reports on it. This is the fact that B2B recombinants are inherently unstable for long term protein production. Once you passage virus past 4 growth cycles in cell culture the yield of protein production falls dramatically. By P8 it is non-existent. This is likely due to the bacterial sequences retain within the virus genome, although nobody has addressed this experimentally. At scientific meetings the phenomenon has been described but there is nothing in the literature to my knowledge. Therefore, all current commercial products are made using bacterial sequence-free vectors.
- Continous protein production in insect cells.
- While the use of stably transformed insect cells is attractive for recombinant protein production, the generally much lower yields make this economically difficult. There is also a problem of genetic stability of these cells as without continued antibiotic (e.g. G418) selection) the level of transgene expression declines. The cited advantage of reduced cell debris in the biomass prior to purification is also over emphasized. If baculovirus-infected cells are harvested in a timely manner then cell culture medium contains few of these contaminants.
- Modified insect cells for enhanced protein production with baculoviruses.
There have been some very clever works performed to modify insect cells for improved recombinant protein production. Although some of these are hard to reproduce from personal experience. While fine in principle, for future commercial production of clinical products they will need to be banked as GMP reagents. Both master and working cell banks have to be made. This is a very expensive process. - Virus contaminants in insect cells.
The authors have completely ignored the nodavirus contamination of Hi5 cells and the rhabdovirus presence in Sf cells. Any cell line used for clinical production of proteins must now be free of these agents. This is the one point that I would insist the authors include in their review. We await with some tension before the FDA find other viruses to be removed from insect cells!
So in summary, I am much in favour of this review. It is broad in its scope and detail. It is an enjoyable read and will prove invaluable for those new to the field and for those with experience who just want to get up to date with current developments.
Finally, correct the spelling of "verification" in Fig.2.
Authors: We sincerely thank you for your kind words regarding our manuscript. We are truly pleased that it was received so positively and that you found it to be an enjoyable read.
Although you kindly left it to our discretion whether to address the comments and suggestions, we greatly appreciated their substantial scientific value and the critical approach, which clearly reflects your deep expertise in the field. For this reason, we decided to address all of the points you raised in a newly added paragraph titled "Limitations and challenges in the implementation of insect expression systems in biopharmaceutical production". We are genuinely grateful for your insightful feedback, which has undoubtedly contributed to improving the quality of our work. We hope the paragraph we added reflects the spirit of your suggestions regarding insect cell lines in the biopharmaceutical industry.
Also, the misspelling in Fig. 2 has been corrected.
Round 2
Reviewer 1 Report
Comments and Suggestions for Authors
The new version of the manuscript has all the answers to my revision.
Reviewer 3 Report
Comments and Suggestions for Authors
All comments have been eliminated.